# Patients' experiences and perceptions of physical restraint in psychiatric care: A descriptive qualitative study at Muhimbili National Hospital

Judith Eladi[1], Sofia Sanga[2], Joel Seme Ambikile[2]*

**1** Muhimbili National Hospital, Mloganzila, Dar es Salaam, Tanzania, **2** Department of Clinical Nursing, Muhimbili University of Health and Allied Sciences, Dar es Salaam, Tanzania

* joelambikile@yahoo.com

## Abstract

Physical restraint in psychiatric care remains controversial due to ethical concerns and the risk for physical and psychological harm. Despite its continued use to manage aggression and maintain safety, evidence on patients' experiences in low-resource settings, including Tanzania, is limited. This study explored patients' experiences and perceptions of physical restraint during psychiatric hospitalization at Muhimbili National Hospital. A qualitative descriptive design was used involving 14 purposively selected adult patients who had experienced physical restraint. Data were analyzed using thematic analysis, and seven themes were identified. Participants described restraint as a complex intervention with both perceived benefits and significant harms. While some acknowledged its role in preventing violence, maintaining ward safety, and supporting treatment during acute agitation, many reported negative experiences, including pain and injury, loss of dignity, humiliation, lack of explanation, ignored requests for assistance, and feelings of coercion or punishment. Limited involvement in decision making and the absence of post-restraint debriefing further undermined trust in healthcare providers. Although some participants accepted restraint as necessary in certain situations, many recommended alternatives, such as verbal de-escalation, attentive communication, pharmacological calming, and close observation. When restraint was unavoidable, patients emphasized the need for humane implementation, including minimal force, short duration, continuous monitoring, and respectful interaction. Overall, physical restraint was experienced as both protective and potentially harmful. The findings highlight the need for context-specific patient-centered approaches that prioritize communication, alternatives to restraint, post-restraint engagement, and staff training to strengthen ethical and rights-based mental health care in Tanzania.

**Data availability statement:** Yes - all data are fully available without restriction; All relevant data are within the paper and its Supporting information files.

**Funding:** This work was supported by the Government of Tanzania through the Ministry of Health (to JE), https://www.moh.go.tz/. The funders had no role in study design, data collection and analysis, decision to publish, or preparation of the manuscript.

**Competing interests:** The authors have declared that no competing interests exist.

## 1. Introduction

Physical restraint is defined as any manual method or physical or mechanical device, material, or equipment that immobilizes a patient or restricts their ability to move their arms, legs, body, or head freely [1]. This includes manual restraint, involving direct physical holding by staff, and mechanical restraint, which specifically refers to the use of devices such as straps, belts, cloth ties, restraint chairs, or specialized beds, to limit a patient's movement in clinical settings [2]. In psychiatric care, physical restraint is commonly employed to prevent harm, manage aggression, agitation, or disorientation, and ensure the safety of both patients and staff. It is typically used in situations involving actual or threatened violence, unclassified behavioral disturbances, or when patients are damaging or threatening to damage property [3]. The use of restraint is also shaped by contextual factors, including limited alternative interventions, escalating behaviors, and clinicians' perceptions of risk, with staff often regarding it as a 'necessary evil,' justified by the unpredictable nature of mental illness and the challenging environments in which they work [4,5].

Globally, the prevalence of physical restraint (defined as the proportion of patients exposed to at least one form of restrictive care practice during the period in which they received inpatient mental health services) among adult psychiatric inpatients ranges from 0.3% % and 54.0%, with the pooled prevalence of approximately 14.4% [6]. The wide variation reflects differences in geographical settings, clinical practices, and reporting methods, and some evidence suggests that the use of restraint has increased in recent years [7–9]. Low- and middle-income countries (LMICs), particularly in sub-Saharan Africa, face additional challenges including inadequate infrastructure, limited training, insufficient staffing, overcrowded wards, and a lack of viable alternatives to restraint [10–14]. Despite its widespread use, physical restraint remains highly controversial, raising concerns about safety, trauma, autonomy, human rights, and broader ethical dilemmas in balancing protection with patients' rights [3]. It has been linked to numerous physical and psychological harms, including pain, injury, loss of control, trauma or retraumatization, and feelings of dehumanization [15,16].

Most research on physical restraint comes from high-income countries. For instance, a study in Switzerland emphasize the risks of restraint use in psychiatric care, noting that patients often perceive restraint as punitive or traumatic, leading to deep-seated mistrust of healthcare providers and hindered therapeutic relationships, which are critical to mental health recovery [16]. Evidence from the African context is more limited, and where available, it primarily reflects the perspectives of healthcare providers and policymakers rather than service users. Studies from Nigeria, Ghana, and Ethiopia indicate that coercive measures, including physical restraint, are often justified as being in the patient's best interest and necessary for safety or achieving clinical outcomes. However, these practices may also contribute to emotional harm, reinforce stigma, and limit patient autonomy [12,17–19].

Patients' experiences and perceptions regarding coercive practices remain underexplored across Africa. In neighbouring Kenya, participants reported that seclusion and restraint were often not patient-friendly, offering some perceived benefits such as

safety, relief, and supportive supervision, but also causing significant harms, including coercion, unmet basic needs, abandonment, mistreatment, and neglect [20]. In Nigeria, coercion was perceived as a necessary evil in severe cases but also as anti-therapeutic to recovery, reinforcing stigma and creating a vicious cycle of abuse. Mechanical restraint in particular was described as dehumanizing and excruciatingly painful [21].

In Tanzania, mental health care remains largely institution-based, with limited community services and a significant shortage of mental health professionals [22,23]. The Mental Health Act (2016) provides a legal framework regulating coercive measures, and Tanzania is a signatory to international human rights instruments, including the UN Convention on the Rights of Persons with Disabilities (UNCRPD), the African Charter on Human and Peoples' Rights (ACHPR), and its disability protocol. Despite these frameworks, socio-cultural beliefs, resource constraints, and workforce limitations may influence both the use and acceptance of physical restraints. Understanding the Tanzanian context is therefore essential to interpreting patients' experiences and developing context-specific, patient-centered recommendations. Given these gaps, this study aimed to explore patients' experiences and perceptions of physical restraint during psychiatric care at Muhimbili National Hospital (MNH). The study sought to generate context-specific qualitative evidence to inform the design of appropriate interventions and strengthen inpatient psychiatric services in Tanzania.

## 2. Materials and methods

### 2.1 Study design

A qualitative descriptive study design was employed to examine the lived experiences and perceptions of physical restraint among patients receiving psychiatric care. This approach enabled an in-depth exploration of patients' subjective meanings, interpretations, and emotional responses within their natural clinical context, while allowing flexibility to identify emerging patterns during analysis, consistent with thematic analysis principles [24].

### 2.2 Study setting

The study was conducted at the Psychiatric Department of MNH in Dar es Salaam region, Tanzania, the country's largest tertiary referral hospital. The department has a total bed capacity of 98 across five wards: male, female, acute, Vikuruti rehabilitation (a structured psychosocial rehabilitation center for patients recovering from mental health crises), and Intramural Private Practice (IPPM) units. MNH provides comprehensive psychiatric and mental health services, including inpatient and outpatient care, methadone-assisted therapy, child and adolescent psychiatry, and rehabilitation services. The site was selected because it serves a large and diverse psychiatric population referred from lower-level health facilities in the region where inpatient psychiatric care is limited, making it an appropriate setting for examining experiences of physical restraint.

### 2.3 Study population

The study population comprised adult psychiatric patients aged 18 years and above who had previously experienced physical restraint during hospitalization at MNH and were attending outpatient follow-up clinics at the time of recruitment. Participants were required to be clinically stable and cognitively able to provide informed consent.

### 2.4 Sampling and sample size

Purposive sampling was used to recruit participants with first-hand experience of physical restraint to ensure information-rich cases. Maximum variation was sought in terms of age, sex, diagnosis, and time elapsed since the most recent restraint episode. Sample size was guided by data saturation. Fourteen participants were interviewed, with thematic saturation reached after twelve interviews and confirmed through two additional interviews [25].

## 2.5  Data collection

Data were collected from 21st April to 30th May 2025 using semi-structured in-depth interviews (IDIs) developed with reference from previous studies conducted in Kenya and Iran [20,26]. The guide included open-ended questions exploring participants' perceptions and experiences before, during, and after episodes of physical restraint. Patients attending the outpatient clinic who had previously experienced physical restraint were identified by a nurse working at the clinic. Immediately after their clinic visit, identified patients were directed to a private room to meet the researcher (PI), who confirmed prior restraint experience, provided study information, and obtained informed consent. A brief mental status evaluation was conducted before the interview, which was then carried out face-to-face in Kiswahili in the same private setting, ensuring confidentiality and participant comfort. The mental status evaluation was conducted to determine participants' capacity to provide informed consent and meaningfully engage in the interview, based on their orientation, understanding of the study information, ability to communicate a choice, and appreciation of the voluntary nature of participation. Only patients who were clinically stable and judged capable of providing informed consent were included in the study. Each interview lasted approximately 30–45 minutes and was audio-recorded with participants' consent. Field notes were taken to document contextual details and non-verbal cues. Participants did not receive any form of compensation.

The PI is a mental health professional affiliated with MNH. To minimize potential power dynamics and social desirability bias, participants were recruited from outpatient clinics where the PI had no direct clinical responsibility. The voluntary nature of participation was emphasized, and interviews were conducted in a private setting. The researchers practiced reflexivity by maintaining field notes and critically reflecting on personal assumptions during data collection and analysis. As supported by research evidence, the researchers hold the position that less restrictive measures should be prioritized in mental healthcare.

## 2.6  Data management and analysis

Audio recordings were transcribed verbatim in Kiswahili, translated into English, and back-translated to ensure linguistic and conceptual accuracy. Data were analyzed manually using Braun and Clarke's six-phase thematic analysis framework [24], including familiarization with the data, initial coding, theme development, theme review, theme definition, and report writing. An inductive and iterative coding process was applied, allowing emerging insights to refine the analytical framework. To enhance dependability and confirmability, three transcripts were independently coded by a senior qualitative researcher and compared with the primary investigator's coding until consensus was achieved. Manual coding was used to facilitate close engagement with the data.

## 2.7  Ethical considerations

Ethical clearance was obtained from the Muhimbili University of Health and Allied Sciences Research and Ethics Committee with reference number MUHAS-REC-03-25-2709. Institutional permission to conduct the study was granted by Muhimbili National Hospital management. To minimize potential power imbalances, the PI had no direct clinical care relationship with the participants, and recruitment was conducted through clinic staff rather than direct solicitation. Participants were informed that participation was entirely voluntary and that their decision to participate or decline would not affect their treatment or relationship with healthcare providers. All participants provided written informed consent after being informed of the study's purpose, procedures, and the right to withdraw at any time without consequences. Interviews were conducted in a private setting to promote comfort and openness. Participant anonymity was ensured through the use of pseudonyms, and all data were securely stored on password-protected devices accessible only to the research team.

## 3.  Results

### 3.1  Socio-demographic characteristics of participants

Participants ranged in age from 23 to 55 years and had diverse educational backgrounds, ranging from primary education to university level. Most participants were unmarried and reported informal employment. All had a history of psychiatric

hospitalization (with primary psychiatric diagnoses being schizophrenia, bipolar disorders, depression, and substance use) and had experienced physical restraint at least once. Detailed socio-demographic characteristics are presented in Table 1.

### 3.2 Emerging themes

Seven major themes emerged regarding participants' experiences and perceptions of physical restraint: (1) Restraint as a Safety and Control Measure, (2) Physical and Psychological Suffering, (3) Loss of Dignity and Basic Human Needs, (4) Poor Communication and Exclusion from Decision-Making, (5) Restraint as Punishment Versus Care, (6) Conditional Trust in Healthcare Providers, and (7) Preference for Alternatives and Humane Practices. Each theme includes sub-themes that capture the depth and complexity of participants' experiences and perceptions. Verbatim quotations, identified by participant number, illustrate these lived experiences. Themes, subthemes, and illustrative codes are presented in Table 2.

**3.2.1 Restraint as a safety and control measure.** This theme describes participants' perceptions of restraint as a necessary intervention to maintain safety, control aggressive behavior, and enable treatment within the psychiatric ward. It encompasses three subthemes including preventing violence and harm, facilitating treatment and medication, and maintaining ward and community order.

***Preventing violence and harm***: Participants in this study perceived restraint as a protective measure to prevent patients from harming themselves, other patients, or staff, as reported by a male participant:

*When a patient is told to sit somewhere and refuses, or hits another patient without cause, that's when restraints are used so they can be given medication and eventually recover and rejoin others.*" (Participant 12)

***Facilitating treatment and medication***: Participants reported the necessary use of restraint to enable administration of medication and continuation of treatment during acute episodes.

*Restraints are used because of human behavior. Some patients are aggressive and refuse treatment. They may need to be restrained to receive medication. It's for their benefit, to help them get back on track.* (Participant 12)

**Table 1. Participants' socio-demographic characteristics.**

| Participant | Age range (Years) | Sex | Marital status | Level of Education | Employment | Treatment duration |
|---|---|---|---|---|---|---|
| P1 | 51-60 | F | Single | Primary | Formal | 1-5 years |
| P2 | 31-40 | M | Single | University | Informal | 1-5 years |
| P3 | 41-50 | M | Separated | Secondary | Formal | >10 years |
| P4 | 41-50 | F | Married | Primary | Informal | ≤1 month |
| P5 | 31-40 | M | Married | Secondary | Formal | 5-10 years |
| P6 | 51-60 | M | Married | Secondary | Informal | >10 years |
| P7 | 21-30 | M | Single | Secondary | Informal | 1-5 years |
| P8 | 31-40 | F | Single | College | Informal | 1-5 years |
| P9 | 31-40 | M | Single | College | Informal | 5-10 years |
| P10 | 41-50 | M | Separated | Secondary | Informal | >10 years |
| P11 | 21-30 | M | Single | Primary | Formal | 1-5 years |
| P12 | 21-30 | M | Widowed | Primary | Informal | ≤1 month |
| P13 | 21-30 | M | Single | University | Formal | 1-5 years |
| P14 | 31-40 | M | Single | Primary | Formal | 1-5 months |

**Table 2. Summary of themes and subthemes on participants' experiences and perceptions of physical restraint.**

| Theme | Subthemes |
|---|---|
| Restraint as a Safety and Control Measure | Preventing violence and harm |
| | Facilitating treatment |
| | Maintaining ward order |
| Physical and Psychological Suffering | Physical pain and injury |
| | Emotional distress |
| Loss of Dignity and Basic Human Needs | Denial of toileting |
| | Humiliation |
| | Prolonged immobilization |
| Poor Communication and Exclusion from Decision-Making | Lack of explanation |
| | Feeling unheard |
| | No debriefing |
| Restraint as Punishment Versus Care | Perceived punishment |
| | Normalization as discipline |
| | Fear-based compliance |
| Conditional Trust in Healthcare Providers | Erosion of trust |
| | Acceptance despite distress |
| | Trust shaped by staff behavior |
| Preference for Alternatives and Humane Practices | Medication and sedation |
| | Verbal de-escalation |
| | Humane restraint |

**Maintaining ward and community order**: The use of restraint was also reported as a necessary means to keep the ward calm and maintain order in the community. It was considered helpful for addressing disturbances, preventing disorder and chaos, monitoring behavior, and maintaining ward control, as described by a male participant:

> If you're disruptive in the ward, if you don't settle, while others are resting, they'll restrain you. … If someone becomes a danger to themselves or others due to lack of calm, they'll be tied up, … in such cases, it's necessary to ensure safety. (Participant 5)

A male participant who was concerned about community safety reported:

> It's the same reason, they (patients) might be sent home with medication and cause chaos or harm. That creates an emergency and adds to the existing problem. That's why patients are restrained, to prevent issues in the community. (Participant 12)

**3.2.2 Physical and psychological suffering.** This theme represents the bodily pain and emotional distress experienced during and after restraint. The theme includes two subthemes: physical pain and injury and emotional and psychological distress.
**Physical pain and injury**: Despite perceived necessity, participants described bodily harm resulting from restraint including swelling, bruises, wounds, scars, tight restraints, impaired circulation, stiffness, heaviness, and fatigue. A female participant verbalized her experience:

> My legs were swollen from being in one position, and my body felt heavy. (Participant 8)

A male participant highlighted the effects of restraint, likening the experience to being crucified:

*The wrists and ankles get bruised. You can't move, like being on a cross.* (Participant 7)

Participants expressed distress related to being restrained for extended periods, resulting in stiffness, heaviness, and fatigue as reported by a male participant:

*They say it (the restraint) is to prevent the patient from falling after sedation. But I don't like it. It makes my heart race, causes fatigue, bruising, and lasts long, from noon until the next morning. It's painful and humiliating.* (Participant 3)

***Emotional and psychological distress***: This subtheme presents emotional reactions to restraint experiences that participants experienced. Restraint was often reported to be associated with the experience of stress, sadness, fear, helplessness, embarrassment, oppression, and loss of happiness as explained by

*My stress increased instead of reducing … because I felt I was being restrained for no reason.* (Participant 9)

A male participant described feeling like a prisoner and experiencing nightmares after being restrained:

*I felt like a prisoner, oppressed. It made me question how bad my condition was to deserve that. …Yes, I had nightmares of being crucified. My mom told me it's because of the restraints.* (Participant 2)

### 3.2.3 Loss of dignity and basic human needs.

This theme captures how restraint compromised patients' dignity, privacy, and access to fundamental needs. Its subthemes include denial of toileting, and humiliation and dehumanization. ***Denial of toileting***: Besides, being restrained, participants in this study described other negative experiences associated with being restrained such as urinating in bed, defecating on self, denied toilet access, and shame. A male participant reported:

*I felt very bad. I needed to use the toilet but was restrained, so I relieved myself in bed.* (Participant 9)

***Humiliation and dehumanization***: This subtheme captures participants' experiences of embarrassment and diminished self-worth resulting from restraint. Participants reported feelings of exposure, being publicly restrained, being treated like a child, and feeling dehumanized. A male participant recounted the negative consequences he experienced as a result of being restrained:

*It hurts. I feel they (those restrained) are being mistreated. I was once restrained and received an injection, and I defecated on myself. It was very embarrassing.* (Participant 11)

### 3.2.4 Poor communication and exclusion from decision-making.

This theme represents participants' experiences of limited explanation, lack of consent, and minimal engagement by healthcare workers. The three subthemes include lack of explanation and consent, feeling unheard or ignored, and absence of post-restraint debriefing. ***Lack of explanation and consent***: This subtheme captures participants' experiences of being restrained without understanding the reasons. Participants reported a lack of explanation, not being informed, assumptions made about them, and resulting confusion. A male participant stated:

*I was tied,... They should have explained to me instead of tying me up and walking me in public like that.* (Participant 3)

In some instances, participants reported that decisions were made unilaterally by staff without obtaining patients' consent:

*It was at night. We'd taken our sleeping medication but hadn't slept. We were chatting in one bed. Staff came in and tied us up, probably to force us to sleep.* (Participant 5)

**Being unheard or ignored:** This subtheme reflects participants' reports of having their requests disregarded. Participants described being ignored when calling for help, having their pleas dismissed, experiencing unmet needs, and being met with silence, as explained by a male participant:

*I felt unsupported. They tied us and left to sleep. Even when I cried out in pain, no one came.... I felt very lonely. I kept calling for help but no one came. I felt discouraged.* (Participant 5)

A male participant who needed to go to the toilet but was ignored reiterated his experience:

*I called the nurse and asked to be released to use the toilet, but they didn't come.* (Participant 10)

**Absence of post-restraint debriefing:** This subtheme describes the absence of follow-up discussions after restraint. Participants reported no post-restraint follow-up, lack of emotional check-ins, and no opportunity for reflection following the restraint. A male participant expressed concern that no healthcare providers inquired about his feelings following the restraint.

*No one asked me about my feelings. … It creates a bit of a gap. No one came to ask about my emotions, it becomes difficult.* (Participant 12)

A female participant expressed concern that she was not informed afterward and reported her wish to have been briefed about the restraint experience:

*I thought they would explain what happened and what to expect. That would have helped me understand better.* (Participant 4)

**3.2.5 Restraint as punishment versus care.** This theme reflects contrasting interpretations of restraint as either therapeutic or punitive, with three subthemes including perceived punishment and mistreatment, normalization of restraint as discipline, and fear-based compliance.

**Perceived punishment and mistreatment:** This subtheme captures experiences in which restraint was perceived as coercive or abusive. Participants described restraint as punishment, mistreatment, cruelty, and an unnecessary intervention.

*Being restrained feels like a punishment. When your mental state returns, the pain is intense, and it acts as a deterrent, you'll want to avoid being restrained again.* … When restrained, you feel like a victim, mistreated and misunderstood (Participant 6)

A male participant who perceived restraint as unnecessary commented:

*I don't believe it's the only solution. They should find other ways. It's not necessary to tie someone up. … tying was unwarranted. It sends the wrong message.* (Participant 3)

**Normalization of restraint as discipline:** This subtheme reflects the normalization of restraint as routine or disciplinary care. Participants described restraint as a means of discipline and control, embedded within hospital rules and everyday practice.

*It (restraint) is not a crime. The hospital is a place for care and discipline*. (Participant 10)

When asked about whether the use of restraint is necessary in some situations, a male participant responded:

*To assess their mental condition, control their aggression, and monitor their behavior.* (Participant 12)

**Fear-based compliance**: This subtheme describes behavior changes driven by fear of repeated restraint. Participants reported modifying their behavior to ensure obedience, silence, and compliance in order to avoid being restrained again.

*You lose trust and start fearing the caregivers after being restrained. … Yes, I lost trust and started strictly following instructions to avoid being tied again.* (Participant 5)

### 3.2.6 Conditional trust in healthcare providers.

This theme describes how restraint experiences shaped participants' trust in healthcare staff. Three subthemes included loss or erosion of trust, acceptance despite distress, and trust influenced by staff behavior.

**Loss or erosion of trust**: This subtheme reflects diminished confidence in caregivers following the administration of restraint. Participants described developing mistrust and resentment, feeling unsafe, and engaging in avoidance of healthcare providers.

*They should know that being restrained puts you in a terrible state. You lose trust and start fearing the caregivers.* (Participant 5)

**Acceptance despite distress**: This subtheme captures continued trust in caregivers despite negative experiences with restraint. Participants described forgiveness, acceptance, normalization of restraint, and an understanding of systemic constraints. A female participant expressed forgiveness toward caregivers following the use of restraint:

*It's hard. You develop some resentment. They say it was for your safety, and you forgive them. But it still hurts.* (Participant 1)

A male participant reiterated the understanding of the use of restraint as part of the process:

*At first, I felt mistreated, but later I understood, it's part of the process. If you're calm, they help you*. (Participant 10)

**Trust influenced by staff behavior**: This subtheme reflects how explanation and gentleness shaped patients' trust in healthcare providers. Participants emphasized respectful care, gentle handling, clear explanations, and reassurance as essential to building trust in healthcare providers, as emphasized by a male participant:

*Don't take things lightly. Be respectful and gentle with patients. Don't restrain unnecessarily, it may drive patients away. … Be honest and explain the situation to the patient, ask for their cooperation, like a police officer would.* (Participant 10)

### 3.2.7 Preference for alternatives and humane practices.

This theme represents participants' recommendations for less coercive and more respectful care approaches. It consists of four subthemes including sedation and medication alternatives, verbal de-escalation and communication, secure observation spaces, and gentle and time-limited restraint.

**Sedation and medication alternatives**: This subtheme reflects participants' preference for pharmacological methods over physical restraint. They suggested using sedatives, calming injections, or medication to manage agitation, as illustrated by a male participant:

*Let patients rest instead of restraining. Use sleeping pills or injections to calm them. Tying should be a last resort.* (Participant 2)

***Verbal de-escalation and communication***: This subtheme highlights the use of dialogue and reassurance to manage patient behavior instead of physical restraint. Participants emphasized techniques such as calm conversation, explanation, persuasion, and attentive listening, as illustrated by a male participant:

*If someone has a problem, talk to them. There's no need to tie them.* (Participant 2)

Another male participant emphasized the importance of listening as a means to manage and control patients' behavior:

*They should listen to the patient, hear what they need and give advice if it's not harmful.* (Participant 9)

***Secure observation spaces***: This subtheme reflects suggestions for safe rooms instead of restraints. Participants suggested the use of observation rooms, secure wards, with close monitoring, instead of restraints as verbalized by a female participant:

*A special room with nothing harmful inside, where the patient can be observed until calm and then returned to the ward.* (Participant 1)

***Gentle and time-limited restraint***: This subtheme reflects calls for minimal and humane use of restraint when unavoidable. Participants emphasized using loose restraints, limiting duration, and monitoring circulation, as reported by a male participant:

*If you must restrain, do it gently, don't cause injury. Blood needs to circulate properly. Restrain moderately and communicate.* (Participant 11)

A male participant who was concerned about the duration of the restraint emphasized:

*Use restraint only for those with aggressive behavior, and don't keep them tied for too long.* (Participant 10)

## 4. Discussion

This study explored the experiences and perceptions of patients receiving psychiatric and mental health care regarding the use of restraints at Muhimbili National Hospital. Seven major themes emerged, reflecting the multifaceted impact of restraint on patients' physical, psychological, and social well-being, as well as their interactions with healthcare providers. These findings provide insights into the benefits, harms, and patient-centered strategies for restraint use in psychiatric care.

Participants reported experiencing a range of physical and psychological consequences following restraint. Physical effects included bruises, restricted movement, and discomfort, while emotional consequences encompassed stress, embarrassment, and feelings of humiliation. These findings align with previous research in Africa and other regions indicating that restraint, although intended for safety, can have unintended negative effects on patients' well-being, dignity, and autonomy [15,16,27–29]. The experience of being restrained often resulted in a sense of vulnerability, emphasizing the importance of careful consideration when implementing such measures [30].

Despite the negative experiences, some participants recognized restraint as necessary for maintaining safety and order within the ward. Restraint was seen as an effective method for preventing harm to other patients and staff, controlling disruptive behavior, and supporting treatment in acute situations. This finding aligns with previous research indicating that

restraint is a necessary evil used as a last resort due to safety concerns [29]. This perception underscores the ethical and clinical challenge of balancing patient safety with respect for autonomy, suggesting that restraint may be justified in limited circumstances but should be used as a last resort [29,30].

Tanzania is a signatory to the UNCRPD and the ACHPR, which obligate the state to protect the dignity, autonomy, and rights of persons with disabilities, including those with mental health conditions. These instruments require freedom from torture, inhuman, or degrading treatment, and promote inclusion and patient-centered care. Participants' experiences of coercion, loss of autonomy, and psychological distress suggest that current restraint practices may, at times, fall short of these obligations. While the Mental Health Act (2016) provides a legal framework for coercive measures, its implementation must align with these international human rights standards. These findings highlight the need for rights-based approaches, including stricter protocols, monitoring, and ethical training for staff.

This study highlights several aspects of restraint that participants found distressing, including loss of dignity, lack of explanation, ignored requests for assistance, and perceived coercion or punishment. These experiences contributed to feelings of powerlessness and dissatisfaction with care. Consistent with prior literature, the findings suggest that restraint, when implemented without clear communication or attention to patient needs, can exacerbate psychological distress and diminish trust in healthcare providers [31–36]. This underscores the importance of creating an environment that ensures safety while preserving dignity [37]. Adequate staffing, clear communication, and structured opportunities for patients to discuss their experiences with restraint may improve the quality of care and foster safer, more supportive environments when restrictive interventions are deemed necessary [38,39].

Trust emerged as a key factor influencing patients' experiences. Participants indicated that respectful, gentle handling and clear explanations enhanced trust in healthcare providers, while perceived negligence or coercion diminished confidence, consistent with previous studies [26,31,32]. Interestingly, some participants reported forgiveness and maintained trust despite negative experiences, highlighting the complex interplay between patient expectations, provider behavior, and relational dynamics in psychiatric care. These findings reinforce the critical role of communication, empathy, and patient-centered care in fostering therapeutic relationships [40,41].

Participants expressed a preference for non-physical interventions, including pharmacological calming strategies, verbal reassurance, and active listening. Such de-escalation techniques align with international recommendations advocating least-restrictive interventions as first-line strategies for behavior management in mental health settings [42–45]. Some suggested alternatives to restraint such as isolation, close observation, and informal coercion are not necessarily perceived as non-coercive by all patients. Informal coercion, including persuasion, inducement, or threats, can exert significant pressure on patients and may still be experienced as coercive despite lacking formal legal status [46]. Moreover, evidence suggests that there is no clear consensus on which interventions are considered least restrictive from the patient perspective, with preferences varying across individuals and contexts [47]. Some patients may perceive seclusion as less intrusive than restraint, while others still experience it as distressing or coercive [33]. When physical restraint was unavoidable, participants suggested measures to minimize harm, including loose restraints, shorter durations, careful monitoring of circulation, and continuous supervision, consistent with best-practices guidelines [48,49].

The study findings have important implications for mental health services in Tanzania. Given the country's largely institution-based mental health system, limited community services, workforce shortages, and evolving mental health legislation, the results highlight the need for context-specific policies and capacity building. Recommended strategies include: (i) rights-based training for staff on ethical and humane restraint practices in line with UNCRPD and ACHPR obligations; (ii) development and implementation of clear protocols, documentation, and monitoring systems for restraint use; (iii) integration of patient-centered approaches, including timely explanation, post-restraint debriefing, and involvement of patients in care decisions where feasible; and (iv) prioritization of alternatives to physical restraint, such as de-escalation techniques, appropriate pharmacological management, and the use of safe observation spaces where available.

Some limitations should be considered. As the study was conducted in a single hospital setting, the transferability of the findings to other contexts may be limited and should be interpreted with caution. In addition, reliance on self-reported experiences may have introduced recall bias. Despite these limitations, the rich qualitative data provide valuable insights into patients' experiences and perspectives and can inform the development of strategies to promote safer and more humane psychiatric care.

## 5. Conclusion

Physical restraint in psychiatric care is a complex intervention with both perceived benefits and significant potential harms. While some patients recognize its role in ensuring safety and ward control, many experience physical, emotional, and relational consequences, which may, at times, conflict with Tanzania's obligations under the UNCRPD and ACHPR. Patient-centered approaches, including clear communication, de-escalation techniques, pharmacological alternatives, humane restraint practices when necessary, and alignment with human rights standards, are essential to improve patient experiences, maintain trust, and uphold ethical and legal obligations in mental health care. Future research should focus on multisite and mixed-methods studies to examine the prevalence, determinants, and outcomes of restraint, evaluate the effectiveness and implementation of restraint-reduction strategies and alternatives in resource-limited settings, and generate comparative evidence across Tanzania and sub-Saharan Africa to inform context-appropriate, rights-based policies and practice.

## Supporting information

**S1 File. Excerpts from interviews.**
(DOCX)

## Author contributions

**Conceptualization:** Judith Eladi.

**Data curation:** Judith Eladi.

**Formal analysis:** Judith Eladi, Sofia Sanga, Joel Seme Ambikile.

**Investigation:** Judith Eladi, Sofia Sanga.

**Methodology:** Judith Eladi, Joel Seme Ambikile.

**Project administration:** Judith Eladi.

**Supervision:** Sofia Sanga, Joel Seme Ambikile.

**Validation:** Sofia Sanga, Joel Seme Ambikile.

**Visualization:** Sofia Sanga, Joel Seme Ambikile.

**Writing – original draft:** Judith Eladi, Joel Seme Ambikile.

**Writing – review & editing:** Judith Eladi, Sofia Sanga, Joel Seme Ambikile.

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
