## [Decision Letter · Decision Letter 0]

4 Feb 2026

PMEN-D-25-00612

Patients’ Experiences and Perceptions of Physical Restraint in Psychiatric Care. A Descriptive Qualitative Study at Muhimbili National Hospital

PLOS Mental Health

Dear Dr. Ambikile,

Thank you for submitting your manuscript to PLOS Mental Health. After careful consideration, we feel that it has merit but does not fully meet PLOS Mental Health’s publication criteria as it currently stands. Therefore, we invite you to submit a revised version of the manuscript that addresses the points raised during the review process.

Please note that we have only been able to secure a single reviewer to assess your manuscript. We are issuing a decision on your manuscript at this point to prevent further delays in the evaluation of your manuscript. Please be aware that the editor who handles your revised manuscript might find it necessary to invite additional reviewers to assess this work once the revised manuscript is submitted. However, we will aim to proceed on the basis of this single review if possible.

Please attend to the expert reviewer's concerns regarding additional contextual details and framing concerns.

We look forward to receiving your revised manuscript.

Kind regards,

Avanti Dey, PhD

Staff Editor

PLOS Mental Health

Journal Requirements:

Additional Editor Comments (if provided):

Reviewers' comments:

Reviewer's Responses to Questions

**Comments to the Author**

1. Does this manuscript meet PLOS Mental Health’s publication criteria? Is the manuscript technically sound, and do the data support the conclusions? The manuscript must describe methodologically and ethically rigorous research with conclusions that are appropriately drawn based on the data presented.

Reviewer #1: Yes

2. Has the statistical analysis been performed appropriately and rigorously?

Reviewer #1: I don't know

3. Have the authors made all data underlying the findings in their manuscript fully available (please refer to the Data Availability Statement at the start of the manuscript PDF file)?

Reviewer #1: No

4. Is the manuscript presented in an intelligible fashion and written in standard English?

Reviewer #1: Yes

5. Review Comments to the Author

Reviewer #1: Thank you for the opportunity to review this manuscript. It is well-written and addresses an important topic that has received limited attention in the study context, making this contribution particularly relevant and timely.

Here are areas for improvement in the manuscript.

1. Page 3, Line 45: How would the authors differentiate the definition of physical restraint from mechanical restraint?

2. Lines 58-60: More recent studies estimate the global prevalence of restrictive measures. See Belayneh et al., 2024; Savage et al., 2024.

3. For a study conducted in a low-resource setting, the authors do not engage sufficiently with literature from low-resource settings. Given the limited number of qualitative studies on this theme conducted on the African continent, including Nigeria and Ghana, I expect them all to be referenced in the manuscript.

4. Other countries on the continent have explored the experiences and perceptions of patients regarding coercive measures, including physical restraints (see Aluh et al., 2022). How will the current study differ from what has been reported?

Consider providing a brief description of the unique context in Tanzania and how it differs from other African countries to give readers a clearer understanding.

5. There should be a description of the authors` positionality and reflexivity. Is the P.I. a staff member of the same hospital?

6. Which ethical safeguards were in place to manage the power dynamics between the P.I. and the study participants?

7. In the abstract, it is written that a framework method of analysis was used, and in the manuscript, a thematic analysis was used. Please clarify.

8. Given that the study aims to explore these experiences in order to provide context-specific recommendations, the recommendation section is not sufficiently explored. A description of the Tanzanian context would allow readers to understand how and why physical restraints are used. Key Contextual Questions to Address include: What is the legal framework? Is there a national mental health Act regulating coercive measures? Is Tanzania a signatory to the UNCRPD, the African Charter on Human and Peoples' Rights (ACHPR), and its disability protocol? How many mental health services and professionals are in the country? Do patients pay out-of-pocket for mental health care? Is there a cultural connotation to the use and acceptance of physical restraints by professionals and patients?

Introduce these details in the background section, then link them with the results in the discussion to make the manuscript richer and more nuanced.

9. What are the future research directions on this theme in Tanzanian mental health care and on the continent?

6. PLOS authors have the option to publish the peer review history of their article (what does this mean?). If published, this will include your full peer review and any attached files.

**Do you want your identity to be public for this peer review?** For information about this choice, including consent withdrawal, please see our Privacy Policy.

Reviewer #1: **Yes:** Deborah Oyine Aluh

Figure Resubmissions:

---

## [Decision Letter · Decision Letter 1]

18 Mar 2026

PMEN-D-25-00612R1

Patients’ Experiences and Perceptions of Physical Restraint in Psychiatric Care. A Descriptive Qualitative Study at Muhimbili National Hospital

PLOS Mental Health

Dear Dr. Ambikile,

Thank you for submitting your manuscript to PLOS Mental Health. After careful consideration, we feel that it has merit but does not fully meet PLOS Mental Health’s publication criteria as it currently stands. Therefore, we invite you to submit a revised version of the manuscript that addresses the points raised during the review process.

Please attend to Reviewer #2's comments requesting greater clarification throughout the manuscript, particularly the results and discussion.

We look forward to receiving your revised manuscript.

Kind regards,

Avanti Dey, PhD

Staff Editor

PLOS Mental Health

Journal Requirements:

Additional Editor Comments (if provided):

Reviewers' comments:

Reviewer's Responses to Questions

**Comments to the Author**

1. If the authors have adequately addressed your comments raised in a previous round of review and you feel that this manuscript is now acceptable for publication, you may indicate that here to bypass the “Comments to the Author” section, enter your conflict of interest statement in the “Confidential to Editor” section, and submit your "Accept" recommendation.

Reviewer #1: All comments have been addressed

Reviewer #2: (No Response)

2. Does this manuscript meet PLOS Mental Health’s publication criteria? Is the manuscript technically sound, and do the data support the conclusions? The manuscript must describe methodologically and ethically rigorous research with conclusions that are appropriately drawn based on the data presented.

Reviewer #1: Yes

Reviewer #2: Yes

3. Has the statistical analysis been performed appropriately and rigorously?

Reviewer #1: N/A

Reviewer #2: N/A

4. Have the authors made all data underlying the findings in their manuscript fully available (please refer to the Data Availability Statement at the start of the manuscript PDF file)?

Reviewer #1: No

Reviewer #2: Yes

5. Is the manuscript presented in an intelligible fashion and written in standard English?

Reviewer #1: Yes

Reviewer #2: Yes

6. Review Comments to the Author

Reviewer #1: Thank you for addressing my review comments.

Reviewer #2: The manuscript presents an interesting, rigorous and well-written study which will add important findings to the literature. I only have minor suggestions for improvement.

Abstract

It would be helpful to include the number of study participants.

Intro

l. 35: What does “the prevalence of physical restraint among adult psychiatric inpatients” refer to? E.g., prevalence of experiencing restraint at least once during their stay?

Methods

l. 169: “recruitment was conducted through clinic staff rather than direct solicitation” - It would be helpful to explain how exactly participants were recruited, e.g. who exactly approached them about participation (treating physicians?) and how/when this was done, and whether participants received any form of compensation for participation in the study.

Regarding reflexivity, it would be helpful to add a statement on the researchers’ own position regarding the use of coercion in mental healthcare, and how this might have influenced the study results.

l.111: It would be helpful to explain the term “Vikuruti rehabilitation” for international readers

Results

I worry whether the detailed description of participants including exact ages, exact treatment duration and occupation risks re-identification of individual patients. I would rather include age ranges (e.g., 20-30 etc.) and ranges of treatment duration, and I am not sure whether information about occupation is relevant to the study – especially given that the text states that most “reported informal employment or were unemployed”. (This seems like a discrepancy between employment information in the table and in the text?) In addition, could the authors provide information about participants’ diagnoses, e.g. bipolar / schizophrenia / etc?

Table 2 is somewhat hard to read. I recommend creating individual rows for the subthemes of each theme, rather than separating subthemes by semicolons. It is unclear which subthemes the illustrative codes correspond to. It would be helpful to either align illustrative codes with the corresponding subthemes or to leave them out.

Discussion

l. 477ff. Patients in this study mentioned several alternatives to restraint, including isolation, observation, and forms of informal coercion. It might be helpful to discuss that these interventions are also perceived as coercive by many patients, so do not necessarily constitute a benign intervention from the perspective of all patients. E.g., the authors might reference literature on informal coercion, and literature on how patients may have different perceptions of which interventions is considered the least restrictive.

Typos/language:

l. 69: Typo: “Evidence from African context is more limited”

l. 76-77: “Patients’ experiences and perceptions regarding coercive practices remain

underexplored across African”

l. 277: “being treated like a child despite adulthood” – I would recommend removing “despite adulthood”, as this is clear from the context

l. 417: “as reported by a male” – I would add “participant” here

7. PLOS authors have the option to publish the peer review history of their article (what does this mean?). If published, this will include your full peer review and any attached files.

**Do you want your identity to be public for this peer review?** For information about this choice, including consent withdrawal, please see our Privacy Policy.

Reviewer #1: **Yes:** Deborah Oyine Aluh

Reviewer #2: No

Figure Resubmissions:

---

## [Decision Letter · Decision Letter 2]

13 Apr 2026

Patients’ Experiences and Perceptions of Physical Restraint in Psychiatric Care. A Descriptive Qualitative Study at Muhimbili National Hospital

PMEN-D-25-00612R2

Dear Dr. Ambikile,

We are pleased to inform you that your manuscript 'Patients’ Experiences and Perceptions of Physical Restraint in Psychiatric Care. A Descriptive Qualitative Study at Muhimbili National Hospital' has been provisionally accepted for publication in PLOS Mental Health.

Best regards,

Karli Montague-Cardoso

Staff Editor

PLOS Mental Health

Reviewer Comments (if any, and for reference):

Reviewer's Responses to Questions

**Comments to the Author**

1. If the authors have adequately addressed your comments raised in a previous round of review and you feel that this manuscript is now acceptable for publication, you may indicate that here to bypass the “Comments to the Author” section, enter your conflict of interest statement in the “Confidential to Editor” section, and submit your "Accept" recommendation.

Reviewer #1: (No Response)

Reviewer #2: All comments have been addressed

2. Does this manuscript meet PLOS Mental Health’s publication criteria? Is the manuscript technically sound, and do the data support the conclusions? The manuscript must describe methodologically and ethically rigorous research with conclusions that are appropriately drawn based on the data presented.

Reviewer #1: (No Response)

Reviewer #2: Yes

3. Has the statistical analysis been performed appropriately and rigorously?

Reviewer #1: (No Response)

Reviewer #2: N/A

4. Have the authors made all data underlying the findings in their manuscript fully available (please refer to the Data Availability Statement at the start of the manuscript PDF file)?

Reviewer #1: (No Response)

Reviewer #2: Yes

5. Is the manuscript presented in an intelligible fashion and written in standard English?

Reviewer #1: (No Response)

Reviewer #2: Yes

6. Review Comments to the Author

Reviewer #1: (No Response)

Reviewer #2: The authors have addressed all reviewer comments. I recommend acceptance for publication.

7. PLOS authors have the option to publish the peer review history of their article (what does this mean?). If published, this will include your full peer review and any attached files.

**Do you want your identity to be public for this peer review?** For information about this choice, including consent withdrawal, please see our Privacy Policy.

Reviewer #1: **Yes:** Deborah Oyine Aluh

Reviewer #2: No
